

# Model ─ TCCON comparisons of column-averaged methane with a focus on the stratosphere

Andreas Ostler[1], Ralf Sussmann[1], Prabir K. Patra[2], Sander Houweling[3,4], Marko De Bruine[3], Gabriele P. Stiller[5], Florian J. Haenel[5], Johannes Plieninger[5], Philippe Bousquet[6,7], Yi Yin[6,7], Marielle Saunois[6,7], Kaley A. Walker[8], Nicholas M. Deutscher[9,10], David W. T. Griffith[9], Thomas Blumenstock[5], Frank Hase[5], Thorsten Warneke[10], Zhiting Wang[10], Rigel Kivi[11], and John Robinson[12]

[1]Karlsruhe Institute of Technology, IMK-IFU, Garmisch-Partenkirchen, Germany
[2]Research Institute for Global Change, JAMSTEC, Yokohama, Japan
[3]Institute for Marine and Atmospheric Research Utrecht, Utrecht University, Utrecht, the Netherlands
[4]SRON Netherlands Institute for Space Research, Utrecht, the Netherlands
[5]Karlsruhe Institute of Technology, IMK-ASF, Karlsruhe, Germany
[6]Laboratoire des Sciences du Climat et de l'Environnement, IPSL-LSCE, CEA-CNRS-UVSQ, UMR8212 91191, France
[7]Université de Versailles Saint Quentin en Yvelines, France
[8]Department of Physics, University of Toronto, Toronto, Canada
[9]School of Chemistry, University of Wollongong, Wollongong, Australia
[10] Institute of Environmental Physics, University of Bremen, Bremen, Germany
[11] Finnish Meteorological Institute, Arctic Research Center, Sodankylä, Finland
[12]Department of Atmospheric Research, National Institute of Water and Atmospheric Research, Wellington, New Zealand

*Correspondence to*: R. Sussmann (ralf.sussmann@kit.edu)

**Abstract.** The distribution of methane ($CH_4$) in the stratosphere can be a major driver of spatial variability in the dry-air column-averaged $CH_4$ mixing ratio ($XCH_4$), which is being measured increasingly for the assessment of $CH_4$ surface emissions. Chemistry-transport models (CTMs) therefore need to simulate the tropospheric and stratospheric fractional columns of $XCH_4$ accurately for estimating surface emissions from $XCH_4$. Simulations from three CTMs are tested against $XCH_4$ observations from the Total Carbon Column Network (TCCON). We analyze how the model-TCCON agreement in $XCH_4$ depends on the model representation of stratospheric $CH_4$ distributions. Model equivalents of TCCON $XCH_4$ are computed with stratospheric $CH_4$ fields from both the model simulations and from satellite-based $CH_4$ distributions from MIPAS (Michelson Interferometer for Passive Atmospheric Sounding) and MIPAS $CH_4$ fields adjusted to ACE-FTS (Atmospheric Chemistry Experiment Fourier Transform Spectrometer) observations. In comparison to simulated model fields we find an improved model-TCCON $XCH_4$ agreement for all models with MIPAS-based stratospheric $CH_4$ fields. For the Atmospheric Chemistry Transport Model (ACTM) the average $XCH_4$ bias is significantly reduced from 38.1 ppb to 13.7 ppb, whereas small improvements are found for the models TM5 (Transport Model, version 5; from 8.7 ppb to 4.3 ppb), and LMDz (Laboratoire de Météorologie Dynamique model with Zooming capability; from 6.8 ppb to 4.3 ppb), respectively. MIPAS stratospheric $CH_4$ fields adjusted to ACE-FTS reduce the average $XCH_4$ bias for ACTM (3.3 ppb), but increase the average $XCH_4$ bias for TM5 (10.8 ppb) and LMDz (20.0 ppb). These findings imply that the range of satellite-based





stratospheric $CH_4$ is insufficient to resolve a possible stratospheric contribution to differences in total column $CH_4$ between TCCON and TM5 or LMDz. Applying transport diagnostics to the models indicates that model-to-model differences in the simulation of stratospheric transport, notably the age of stratospheric air, can largely explain the inter-model spread in stratospheric $CH_4$ and, hence, its contribution to $XCH_4$. This implies that there is a need to better understand the impact of

individual model transport components (e.g., physical parameterization, meteorological data sets, model horizontal/vertical resolution) on modeled stratospheric $CH_4$.

## 1 Introduction

The column-averaged dry-air mixing ratio of methane ($CH_4$), denoted as $XCH_4$, is an integrated measure of $CH_4$ with contributions from the troposphere and the stratosphere. Observations of $XCH_4$ contain source/sink information on a global

to regional scale. They are provided by the ground-based networks NDACC (Network for the Detection of Atmospheric Composition Change, http://www.ndacc.org/; Kurylo, 1991) and TCCON (Total Carbon Column Observing Network, http://www.tccon.caltech.edu/; Wunch et al., 2011a), and also by satellite-based observation platforms like SCIAMACHY (Scanning Imaging Absorption Spectrometer for Atmospheric Cartography; Burrows et al., 1995; Frankenberg et al., 2011) and GOSAT (Greenhouse Gases Observing Satellite; Kuze et al., 2009; Yokota et al., 2009). Satellite-inferred $XCH_4$

observations are increasingly used in atmospheric inverse modelling because of their beneficial spatiotemporal data coverage (Bergamaschi et al., 2013; Fraser et al., 2013; Monteil et al., 2013; Fraser et al., 2014, Houweling et al., 2014; Wecht et al., 2014; Cressot et al., 2014; Alexe et al., 2015; Turner et al., 2015; Locatelli et al., 2015). Given the high accuracy of ground-based $XCH_4$ TCCON retrievals, these observations are typically used for the evaluation of both chemistry-transport model (CTM) simulations (Saito et al., 2012; Belikov et al., 2013; Monteil et al., 2013; Fraser et al., 2014; Alexe et al., 2015;

Turner et al., 2015), and satellite-retrieved $XCH_4$ (Parker et al., 2011; Schepers et al., 2012; Dils et al., 2014; Houweling et al., 2014; Parker et al., 2015; Kulawik et al., 2015; Parker et al., 2015; Pandey et al., 2016; Inoue et al., 2016).

Because of the various influences on $XCH_4$, however, the interpretation of residual $XCH_4$ differences with TCCON may be difficult. For example, a good agreement between $XCH_4$ simulations and observations may suggest that a CTM is able to represent atmospheric conditions in a realistic way. However, it could also be that systematic model and satellite data errors

in the troposphere and the stratosphere compensate each other. For this reason, it is necessary to extend model validations with additional atmospheric $CH_4$ observations that are complementary to $XCH_4$ observations, like surface or airborne in situ measurements, or balloon-based vertical profiles (Karion et al., 2010). In the context of a refined model comparison, it is also possible to separate ground-based $XCH_4$ observations into tropospheric and stratospheric partial columns (Washenfelder et al., 2003; Sepúlveda et al., 2012; 2014; Wang et al., 2014; Saad et al., 2014).

Model-measurement $XCH_4$ residuals are minimized by atmospheric inversions in order to constrain $CH_4$ emission fluxes. Inversion models are also able to make use of in situ measurements and $XCH_4$ observations at the same time in order to adjust prior emission fluxes. Nevertheless, such inverse models still have to deal with ill-defined $XCH_4$ residual biases,



which, in contrast to well-quantified biases, cannot be attributed to errors in the model or the observations without an unambiguous assignment (Houweling et al., 2014). Currently, there are various approaches to optimize bias functions within the inverse model or to construct bias corrections as ad hoc functions of latitude or air mass. Ad hoc bias corrections, like removing a latitudinal background pattern in $XCH_4$ model-observation differences, are common, even though they bear the risk of obscuring real signals from emissions on the Earth's surface. Given the fact that the stratospheric contribution relative to the $CH_4$ total column increases from ~5% at the tropics up to ~25% at mid- and high latitudes, model errors in the representation of stratospheric $CH_4$ mixing ratios are expected to give rise to a latitudinal varying bias (Turner et al., 2015). Although it is known that CTMs differ by up to ~50% in the simulation of lower stratospheric $CH_4$ distributions (Patra et al., 2011), an atmospheric region with a steep methane gradient of ~ -50 ppb/km, the impact of model errors in stratospheric $CH_4$ on $XCH_4$ has not been rigorously quantified up to now. In this context, the goal of this study is to better understand the sensitivity of $XCH_4$ model-observation differences to the model representation of stratospheric $CH_4$.

Our $XCH_4$ model-observation analysis is based on optimized model simulations from three well-established CTMs on the one side and accurate $XCH_4$ observations from TCCON on the other. The impact of model stratospheric $CH_4$ distributions on $XCH_4$ is estimated by replacing modeled stratospheric $CH_4$ fields with monthly mean $CH_4$ distributions observed by MIPAS (Michelson Interferometer for Passive Atmospheric Sounding), and by ACE-FTS (Atmospheric Chemistry Experiment Fourier Transform Spectrometer). In addition to this, we briefly evaluate the model characteristics of stratospheric transport in order to understand differences between simulated and observed $CH_4$ distributions. The paper has the following structure: After introducing the models (Sect. 2) and the observations (Sect. 3), we present both a direct model-TCCON comparison and a comparison with refined model data using satellite data products of stratospheric $CH_4$ in Sect. 4. The transport characteristics of the models are discussed in Sect. 5, followed by a summary and conclusions in Sect. 6.

## 2 Model simulations

The focus of this study is assessing the impact of stratospheric $CH_4$ on $XCH_4$. Therefore, we try to ensure that model simulations represent tropospheric $CH_4$ mixing ratios as well as possible. For this purpose, we use optimized $CH_4$ model simulations that have been constrained by surface observations. Our model analysis comprises simulations from three well-established CTMs that have already been part of the chemistry-transport model inter-comparison experiment TransCom-$CH_4$ (Patra et al., 2011) and used in inverse modelling of $CH_4$ emissions. Furthermore, we use model simulations of stratospheric mean age for an evaluation of model transport characteristics in Sect. 5. Basic model features are given in Table 1.

### 2.1 ACTM

The ACTM model (Patra et al., 2009a) is an atmospheric general circulation model (AGCM)-based CTM from the Center for Climate System Research/National Institute for Environmental Studies/Frontier Research Center for Global Change (CCSR/NIES/FRCGC). Here, we use optimized ACTM simulations presented in Patra et al. (2016) as inversion case 2



(CH4ags). The ACTM horizontal resolution is ~ 2.8°×2.8° (T42 spectral truncations) with 67 sigma-pressure vertical levels. The meteorological fields of ACTM are nudged with reanalysis data from the Japan Meteorological Agency, version JRA-25 (Onogi et al., 2007). ACTM uses an optimized OH field (Patra et al., 2014) based on a scaled version of the seasonally varying OH field from Spivakovski et al. (2000). The concentration fields being relevant for stratospheric CH$_4$ loss − OH,

O($^1$D), and chlorine (Cl) radicals – are based on simulations by the ACTM's stratospheric model run (Takigawa et al., 1999). ACTM mean age is derived from the simulation of an idealized transport tracer with uniform surface fluxes, linearly increasing trend, and no loss in the atmosphere (Patra et al., 2009b). The ACTM simulate the observed CH$_4$ inter-hemispheric gradient in the troposphere and individual in situ measurements generally within 10 ppb (Patra et al., 2016).

## 2.2 TM5

The global chemistry Tracer Model, version 5 (TM5) has been described in Krol et al. (2005) and used as an atmospheric inversion model for CH$_4$ emissions (Bergamaschi et al., 2005; Meirink et al., 2008, Houweling et al., 2014). Here, we use TM5 simulations of CH$_4$ optimized with surface measurements only (Pandey et al., 2016). TM5 is run with a horizontal resolution of 6°×4° and a vertical grid of 25 layers. TM5 meteorology is driven by the reanalysis data set ERA-interim (Dee et al., 2011) from the European Centre for Medium Range Weather Forecasts (ECMWF). The simulation of the chemical

CH$_4$ sink uses OH fields from Spivakovski et al. (2000), which have been scaled to match methyl chloroform measurements. In addition to that, stratospheric CH$_4$ loss via Cl and O($^1$D) radicals is simulated using their concentration fields based on the 2-D photochemical Max-Planck-Institute (MPI) model (Bruehl and Crutzen, 1993). Known deficiencies in the TM5 simulation of inter-hemispheric mixing have been corrected by extending the model with a horizontal diffusion parameterization that is adjusted to match SF$_6$ simulations with SF$_6$ measurements (Monteil et al., 2013).

TM5 simulations of sulfur hexafluoride (SF$_6$) were used to derive stratospheric mean age data. SF$_6$ mixing ratios are monotonically increasing with time showing higher mixing ratios in the troposphere than in the stratosphere, given the transport time from SF$_6$ surface sources to higher altitudes. This implies that tropospheric and stratospheric SF$_6$ mixing ratios of equal size are separated from each other by a time lag which is commonly defined as mean age of air. In order to derive mean age from SF$_6$ model simulations, the same tropospheric SF$_6$ reference time series was used as for the derivation of

MIPAS mean age data (see Stiller et al., 2012)

## 2.3 LMDz

The LMDz (Laboratoire de Météorologie Dynamique model with Zooming capability) is a general circulation model (Hourdin et al., 2006), that has been used to investigate the impact of transport model errors on inverted CH$_4$ emissions (Locatelli et al., 2013). Here, we use optimized LMDz simulations of CH$_4$, recently presented as LMDz-SP constrained by

surface measurements from background sites (Locatelli et al., 2015). These model simulations are nudged with the ERA-Interim reanalysis data set for horizontal winds (u,v). LMDz has a horizontal resolution of 3.75°×1.875°, and 39 hybrid sigma-pressure layers. The chemical destruction of CH$_4$ by OH and O($^1$D) is based on prescribed concentration fields



simulated by the chemistry–climate model LMDz-INCA (Szopa et al., 2013). No Cl-based $CH_4$ destruction is prescribed in this version of the model. Besides $CH_4$, LMDz simulations of $SF_6$ were used to derive mean age data in analogy to the method used for TM5.

## 3 Intercomparison strategy and observations

### 3.1 Intercomparison strategy

We want to quantify the dependence of the $XCH_4$ model-observation agreement on the model representation of stratospheric $CH_4$ mixing ratios. For this purpose, we apply original $CH_4$ model fields and two corrected $CH_4$ model fields, where we have replaced the modeled stratospheric $CH_4$ by satellite data sets of stratospheric $CH_4$ mixing ratios. The first satellite data set consists of MIPAS $CH_4$ observations, whereas the second satellite data set contains MIPAS $CH_4$ observations that are adjusted to ACE-FTS-observed $CH_4$ levels. This allows us to represent an uncertainty range for the satellite-based model correction. Finally, our $XCH_4$ model-observation comparison deals with a triplet of model $CH_4$ fields for each CTM.

Using TCCON $XCH_4$ observations as validation reference, we evaluate the impact of correcting the modeled stratospheric $CH_4$ on $XCH_4$. Consequently, modeled vertical profiles of $CH_4$ were extracted for each TCCON site and subsequently converted to $XCH_4$ by accounting for the TCCON retrieval a priori and vertical sensitivity. This means that model $CH_4$ profiles are adjusted to the actual surface pressure measured at the time of a single TCCON observation. In addition to that, model profiles are convolved with the daily TCCON retrieval a priori profiles of $CH_4$, that have been converted from wet-air into dry-air units by subtracting a daily water vapour profile provided by NCEP (National Centers for Environmental Prediction).and the averaging kernel depending on the actual solar zenith angle. Thereby, monthly mean $CH_4$ profiles from LMDz also receive a daily component depending on the surface pressure, the TCCON a priori profiles and averaging kernels. The statistical analysis of $XCH_4$ model-TCCON differences then is based on the daily mean time series for the year 2010 and produces two site-specific parameters: the mean difference (bias) and the residual standard deviation (RSD).

### 3.2 TCCON observations of column-averaged methane

Solar absorption measurements in the near-infrared (NIR) are performed via ground-based Fourier Transform Spectrometers (FTS) at TCCON sites across the globe. TCCON-type measurements are analyzed with the GGG software package including the spectral fitting code GFIT to derive total column abundances of several trace gases (Wunch et al., 2011). The $CH_4$ total column is inverted from the spectra in three different spectral windows centered at 5938 cm$^{-1}$, 6002 cm$^{-1}$, and 6076 cm$^{-1}$. The spectral fitting method is based on iteratively scaling a priori profiles to provide the best fit to the measured spectrum. The general shape of the a priori profiles has been inferred from aircraft, balloon and satellite profiles (ACE-FTS profiles measured in the 30-40° N latitude range from 2003 to 2007). In addition, the shape of the daily a priori profile is vertically squeezed/stretched depending on tropopause altitude and the latitude of the measurement site. This means, that the tropopause altitude is used as a proxy for stratospheric ascent/descent to represent the origin of the airmass in the a priori



profile. XCH$_4$ is calculated by dividing the CH$_4$ number density by the simultaneously measured O$_2$ number density (a proxy for the dry-air pressure column).

These XCH$_4$ retrievals are a posteriori corrected for known airmass-dependent biases and calibrated to account for airmass-independent biases, which can, among other errors, arise from spectroscopic uncertainties (Wunch et al., 2011). The airmass-independent calibration factor, which is determined by comparisons with coincident airborne or balloon-borne in situ measurements over TCCON sites (Wunch et al., 2010; Messerschmidt et al., 2011; Geibel et al., 2012), allows for a calibration of TCCON XCH$_4$ retrievals to in situ measurements on the WMO scale. Furthermore, the quality of the retrievals is continuously improved by correcting the influence of systematic instrumental changes over time. As a result of these improvements there are different versions of the GGG software package. In this study we use TCCON retrievals performed with version GGG2014 (for details see https://tccon-wiki.caltech.edu/). The TCCON measurement precision (2-$\sigma$) for XCH$_4$ is <0.3% (< 5ppb) for single measurements. For the year 2010, XCH$_4$ observations are available from 11 TCCON sites, listed in Table 2. Knowing that TCCON XCH$_4$ accuracy can be affected by a strong polar vortex (Ostler et al., 2014), we exclude high-latitude observations at Sodankylä within the early spring period (March, April, May) from the analysis. TCCON data were obtained from the TCCON Data Archive, hosted by the Carbon Dioxide Information Analysis Center (CDIAC: http://cdiac.ornl.gov/). The individual data sets of the TCCON sites used in this study are available at this database.

### 3.3 Satellite-based data sets of stratospheric methane

In order to correct modeled stratospheric CH$_4$ fields, we use satellite-borne MIPAS measurements covering the stratosphere. As a Fourier-Transform Infrared Spectrometer aboard the Environmental Satellite (Envisat), MIPAS detected atmospheric emission spectra in the mid-infrared region via limb sounding (Fischer et al., 2008). Profiles of various atmospheric trace gas concentrations are derived by the research processor developed by the Karlsruhe Institute of Technology, Institute of Meteorology and Climate Research (KIT IMK) and the Instituto de Astrofísica de Andalucía (CSIC) (von Clarmann et al., 2003). The MIPAS CH$_4$ data set comprises zonal monthly means with a horizontal grid resolution of 5° latitude. In the vertical, the resolution of the MIPAS CH$_4$ fields range from 2.5 to 7 km, see Plieninger et al. (2015a) for more details. As an additional quality criterion, we only select MIPAS data points that are averaged over more than 300 profile measurements. As a result, our MIPAS CH$_4$ data set typically covers altitudes higher than ~10 km at mid latitudes and heights above ~15 km in the Tropics. This implies that we do not use a thermal or chemical tropopause definition, but use the MIPAS data where they are available. Therefore, we cannot exclude that our MIPAS-based CH$_4$ fields contain some upper tropospheric MIPAS values, i.e. our definition of stratospheric CH$_4$ is not strict from a meteorological point of view.

The corrected model CH$_4$ profiles rely on original model CH$_4$ fields that are merged with MIPAS-based zonal CH$_4$ fields (monthly means) interpolated to the model grid. Merging original model CH$_4$ fields/profiles with zonal monthly means implies that we lose some spatial and temporal variability in the corrected model CH$_4$ fields. However, for our aim ─ investigating the overall impact of model stratospheric CH$_4$ fields on the quantity XCH$_4$ ─ a monthly mean representation of stratospheric CH$_4$ in the corrected model fields is sufficient.



In our study we use the strongly revised MIPAS $CH_4$ data product for the MIPAS reduced-resolution period from January 2005 to April 2012. This new data set (version V5R_CH4_224/V5R_CH4_225) was recently introduced by Plieninger et al. (2015) with an emphasis on retrieval characteristics. Plieninger et al. (2015) showed that $CH_4$ mixing ratios are reduced in the lowermost stratosphere when using the new retrieval settings. This finding implies that the high bias of the older $CH_4$

data version in the lowermost stratosphere, which was determined by Laeng et al. (2015), has been partially alleviated. Nevertheless, a recent comparison study by Plieninger et al. (2016) suggests a remaining positive bias (100 – 200 ppb) relative to other satellite measurements such as ACE-FTS observations.

For this reason, a second satellite $CH_4$ data set was constructed by adjusting MIPAS stratospheric $CH_4$ mixing ratios to ACE-FTS measurements of $CH_4$. Given the sparse data coverage of ACE-FTS observations for the year 2010, we did not use

ACE-FTS measurements directly. Instead, the MIPAS $CH_4$ fields were adjusted by offsets relative to ACE shown in Fig. 1, yielding the second satellite-based $CH_4$ data set abbreviated by MIPAS_ACE. We used collocated pairs of $CH_4$ profiles from MIPAS and ACE-FTS to derive a $CH_4$ offset as a function of altitude and latitude for the year 2010. The collocation criteria are based on a maximum radius of 500 km and a maximum temporal deviation of 5 hours, which is identical to Plieninger et al. (2016). Furthermore, the MIPAS averaging kernels were applied to ACE-FTS $CH_4$ profiles. ACE-FTS operates in solar

occultation mode (Bernath et al., 2005) and also provides retrievals of several trace gases including $CH_4$. Here, we use ACE-FTS data from a research version of the 3.5 retrieval described in Buzan et al. (2015).

Figure 1 shows the $CH_4$ offset functions computed as mean differences between MIPAS and ACE-FTS for 30° latitudinal bands. Figure 1 confirms the findings by Plieninger et al. (2016) that MIPAS is biased positive by ~ 150 ppb relative to ACE-FTS within the lowermost stratosphere. For higher altitudes (> 25 km), mean differences between MIPAS and ACE-

FTS are larger for the tropical domain (up to 100 ppb) compared to higher latitudes (up to 50 ppb).

### 3.4 MIPAS-observed mean age

Besides MIPAS $CH_4$ observations, we also use MIPAS data sets of stratospheric mean age inferred from $SF_6$ measurements. Here, we use the new MIPAS mean age data set presented by Haenel et al. (2015). This new mean age data set contains several improvements compared to the previous version introduced by Stiller et al. (2012). For MIPAS, the mean age is

calculated as the average transport time from the tropical troposphere to a certain location in the stratosphere using NOAA (National Oceanic and Atmospheric Administration) observations as reference. The mean age of stratospheric air is of special interest for climate research because the distributions of greenhouse gases like ozone critically depend on possible changes in the stratospheric transport pathways (Engel et al., 2009). Mean age can be inferred from observations of clock-tracers (concentrations monotonically increasing with time) like $SF_6$ or $CO_2$, and can also be simulated by models. For this

reason, it is a well-known diagnostic for stratospheric transport being very suitable for the evaluation of model transport characteristics (Waugh and Hall, 2002). The combined MIPAS data set of stratospheric $CH_4$ and mean age is used for the evaluation of model transport characteristics in Sect. 5.1.



## 4 Model-TCCON comparison of column-averaged methane

Figure 2 shows model biases in $XCH_4$ with respect to TCCON observations, where each TCCON site is represented by its geographical latitude. For each CTM a triplet of model $CH_4$ fields (uncorrected, MIPAS and MIPAS_ACE corrected) yields a triplet of model $XCH_4$ biases. All site-specific $XCH_4$ model biases are individually listed in Table 3. In addition, Table 4

provides an average $XCH_4$ bias for each model data set, computed as the mean of absolute site-specific biases.

The original $XCH_4$ bias for ACTM lies in between 18.8 ppb and 51.3 ppb (see Fig. 2a and Table 3). This high bias is significantly reduced when ACTM stratospheric $CH_4$ fields are replaced by satellite-based $CH_4$ fields. The model correction with MIPAS $CH_4$ reduces the average ACTM $XCH_4$ bias from 38.1 ppb to 13.7 ppb (see Table 4). Site-specific $XCH_4$ biases are ranging from 4.8 ppb to 19.9 ppb (see Table 3). The model correction with MIPAS_ACE reduces the average ACTM

$XCH_4$ bias further from 38.1 ppb to 3.3 ppb (see Table 4) with values in an interval between −9.9 ppb and 3.5 ppb (see Table 3) ), similar to that were expected from the comparison with ACTM simulations with tropospheric measurements (Patra et al., 2016).

For the original TM5 we detect negative site-specific $XCH_4$ biases with values between −17.6 ppb and −3.7 ppb (see Fig. 2b and Table 3). When TM5 $CH_4$ fields are corrected with MIPAS observations, this negative $XCH_4$ bias is reduced from -8.7

ppb to -4.3 ppb on average (see Table 3). The corresponding site-specific $XCH_4$ biases then are between −11.1 ppb and 8.1 ppb (Table 3). If the MIPAS_ACE is applied to TM5 then the site-specific TM5 $XCH_4$ biases are shifted further to the negative direction with values between −18.3 ppb and −3.7 ppb. In this case the average $XCH_4$ bias increased from 8.7 ppb to 10.8 ppb (Table 4).

With respect to TCCON observations LMDz produces both negative and positive $XCH_4$ biases ranging from −11.9 ppb

(Wollongong) to 13.0 ppb (Sodankylä), see Fig. 2c and Table 3. The average LMDz $XCH_4$ bias is slightly reduced from 6.8 ppb to 4.3 ppb if LMDz is corrected with MIPAS $CH_4$ fields (see Table 4). After this correction, site-specific LMDz $XCH_4$ biases lie between −2.9 ppb and 9.1 ppb. Using MIPAS_ACE $CH_4$ fields for the LMDz model correction produces LMDz $XCH_4$ biases between −13.8 ppb and −31.1 ppb. At the same time, the average LMDz $XCH_4$ bias is increased from 6.8 ppb to 20.0 ppb (Table 4).

Overall, our results confirm that the model-TCCON agreement in $XCH_4$ depends very much on the model representation of stratospheric $CH_4$. It is obvious that the $XCH_4$ offset between ACTM and TCCON is significantly reduced with stratospheric $CH_4$ fields based on satellite data. By contrast, for TM5 and LMDz the impact of the model correction on the model-TCCON agreement is ambiguous. In that, the model-TCCON agreement can be improved (with MIPAS), but can also be reduced (with MIPAS_ACE). In order to understand this inter-model spread we look at the differences between modeled and

satellite-retrieved $CH_4$ fields. Figure 3 shows zonal and annual averaged $CH_4$ mixing ratio differences between MIPAS and each CTM. Figure 3a illustrates that stratospheric $CH_4$ mixing ratios are generally much higher in ACTM than in MIPAS. The ACTM-MIPAS differences in $CH_4$ are increasing from negligible values within the lowermost stratosphere up to 450 ppb in the upper stratosphere. Furthermore, the ACTM-MIPAS difference in $CH_4$ also shows a latitudinal dependence, with




middle and upper stratospheric values increasing towards higher latitudes. The positive bias in stratospheric ACTM $CH_4$ mixing ratios causes a positive ACTM bias in $XCH_4$. In contrast to that, we find negative model-MIPAS differences in stratospheric $CH_4$ mixing ratios for TM5 (Fig. 3b) resulting in a small negative $XCH_4$ bias. We identify two altitude regions, where TM5 modeled $CH_4$ mixing ratios are smaller than MIPAS $CH_4$ mixing ratios: the lower stratosphere with differences

in $CH_4$ mixing ratios of up to $-100$ ppb, and the upper stratosphere ($> 30$ hPa) with maximum $CH_4$ differences of $\sim -150$ ppb. Figure 3c shows the $CH_4$ mixing ratio differences between LMDz and MIPAS with noticeable negative $CH_4$ differences of up to $-200$ ppb within the tropical upper stratosphere. Negative $CH_4$ differences ($\sim -100$ ppb) are also visible in the upper stratosphere of the mid- and high-latitude region. In contrast to this, we identify positive $CH_4$ differences of up to 100 ppb within the middle stratosphere ($\sim 50$ hPa) of the mid and high latitudes. The negative and positive $CH_4$ differences partially

cancel out in $XCH_4$. In analogy to Fig. 3, the $CH_4$ differences between model and MIPAS_ACE fields are illustrated in Fig. 4. Given the offset adjustment of MIPAS to ACE-FTS (see Fig. 1), the MIPAS_ACE $CH_4$ fields comprise lower $CH_4$ mixing ratios compared to MIPAS, mostly in the lower stratosphere. Hence, the ACTM-satellite $CH_4$ difference is larger for MIPAS_ACE fields than for MIPAS fields. For TM5 and LMDz model-satellite $CH_4$ differences are shifted into the positive direction (Figs. 4b and 4c). In other words, modeled stratospheric $CH_4$ mixing ratios appear to be too high when compared to

MIPAS and too low in comparison to MIPAS_ACE.

## 5 Discussion

Our analysis shows that the model-TCCON agreement in $XCH_4$ critically depends on the model representation of stratospheric $CH_4$, which is diverse for the presented CTMs. In the following we discuss possible causes for the inter-model spread in stratospheric $CH_4$. In addition to that, we evaluate the findings of our $XCH_4$ model-TCCON comparison with

respect to satellite data uncertainty.

### 5.1 Model transport characteristics as possible cause for inter-model spread in stratospheric methane

An inter-model spread in stratospheric $CH_4$ fields has already been detected by Patra et al. (2011) despite applying uniform fields of OH, Cl, and $O^1D$ for all models. Their findings, therefore, suggested a predominant role of transport in the simulation of $CH_4$ vertical distributions. For this reason, we tested here whether differences in the modeling of stratospheric

transport are noticeable. To do this, we follow the approach of Strahan et al. (2011) who sought to understand chemistry climate model ozone simulations using transport diagnostics. This method is based on the compact relationship between a long-lived stratospheric tracer and mean age in the lower stratosphere. In their work, they compared simulations and airborne observations of $N_2O$/mean age correlations, in order to evaluate the model transport characteristics. Here, we use the MIPAS data of $CH_4$ and mean age as a reference to identify model-to-model differences in the simulation of stratospheric

transport. The MIPAS data are not used to evaluate, whether modeled stratospheric circulations are realistic or not, given the uncertainties of MIPAS $CH_4$ and mean age data. For example, the MIPAS mean age range may be too large, because MIPAS





mean age can be up to 0.8 years too old due to the impact of mesospheric $SF_6$ loss (Stiller et al., 2012). This loss process was not included in the models used for this study. Moreover, the MIPAS $CH_4$ data significantly differs from ACE-FTS $CH_4$ data within the lower stratosphere (see Fig. 1).

In analogy to Strahan et al. (2011) we focus our model transport diagnostics on the tropical domain because tropical

diagnostics quantities allow a better assessment of the individual transport processes ascent and mixing. Annual means of age and $CH_4$ mixing ratios for modeled as well as MIPAS-observed fields were calculated for the lower stratosphere (30−100 hPa) of the tropical domain (10°S−10°N), and of the northern-hemispheric mid-latitude region (35°N−50°N), respectively. Subsequently, vertical profiles of mean model-MIPAS differences were calculated to provide insight into the tropical transport characteristics.

Figure 5 illustrates that the model-MIPAS difference of tropical mean age is almost identical for all models. I.e. the model simulations produce similar mean ages that are younger than MIPAS-observed mean ages. Knowing that mean age only represents the combined effects of ascent and mixing, we separately look at those two processes being relevant for stratospheric transport. According to Strahan et al. (2011), the tropical ascent rate is assessed by the horizontal mean age gradient, calculated as the difference between mid-latitude and tropical mean ages. The model-MIPAS difference of the

tropical ascent rate is shown in Fig. 6, indicating that ACTM and LMDz simulate tropical ascent in similar way. The TM5-modeled tropical ascent is faster compared to ACTM and LMDz. Next, we look at the tropical model-MIPAS $CH_4$ difference, which is used as a measure for (cumulative) horizontal mixing. Figure 7 reveals that horizontal mixing is different for ACTM compared to TM5 and LMDz looking very similar. I.e. the horizontal mixing appears to be weaker for ACTM compared to the other models. Finally, these model transport diagnostics indicate model-to-model differences in the

simulation of tropical ascent and horizontal mixing, which are likely to cause an inter-model spread in model stratospheric $CH_4$ fields.

Indeed, model-to-model differences affecting the simulation of stratospheric transport are present in the vertical/horizontal resolution, sub-grid-scale physical parameterizations, advection schemes, numerical methods, etc. Furthermore, the simulation of stratospheric transport depends on the reanalysis data used to drive the model meteorology,. e.g. the ECMWF

reanalysis data set ERA-Interim leads to an improved representation of the stratospheric circulation in comparison to the older ERA-40 reanalysis data (Monge-Sanz et al., 2007, 2011; Diallo et al., 2012). The ERA-Interim data are used by TM5 and LMDz, whereas ACTM applies the JRA-25 reanalysis data (Onogi et al., 2007), which is known to have several deficiencies compared to the newer JRA-55 data (Ebita et al., 2011). However, testing ACTM with both ERA-interim/40 and JRA-25/55 has not produced significant differences in $CH_4$ simulations (P. Patra, personal communication, 2016). Besides

that, we do not expect that the poor representation of stratospheric $CH_4$ by ACTM (with 67 vertical levels) is impacted by a coarse vertical model grid resolution, as seen for an older version of LMDz (Locatelli et al., 2015).





## 5.2 Significance of satellite data range

The model correction with satellite-based $CH_4$ fields has an impact on the $XCH_4$ model-TCCON agreement, but the significance of this impact is diverse for the models. For ACTM both satellite-based $CH_4$ fields, in particular MIPAS_ACE, clearly yield an improved model-TCCON agreement. For TM5 and LMDz, the model-TCCON agreement can be slightly improved (with MIPAS), but also reduced (with MIPAS_ACE). Thereby, we assert, that original $XCH_4$ simulations from TM5 and LMDz lie inside the range that is spanned by the two satellite-based $CH_4$ fields. The most prominent feature of the satellite data range lies within the lower stratosphere where MIPAS-retrieved $CH_4$ mixing ratios are up to 200 ppb higher than ACE-FTS-retrieved $CH_4$ mixing ratios. Plieninger et al. (2016) also found a similar high bias for MIPAS $CH_4$ data in comparison to satellite-based $CH_4$ observations from SCIAMACHY or HALOE (HALogen Occultation Experiment). Furthermore, they showed that surface measurements provide $CH_4$ mixing ratios with slightly lower values than MIPAS-retrieved $CH_4$ mixing ratios of the upper troposphere, a finding that is against expectation. For these reasons, it is likely that our satellite data range is dominated by high biased lower stratospheric MIPAS $CH_4$ data. Thus, the model correction with ACE-FTS-based $CH_4$ fields seems more reliable. However, a definite assessment of the satellite data accuracies is not possible yet due to the lack of an extensive observational data set based on stratospheric in situ measurements.

## 6 Summary and conclusions

This study analyzed the importance of uncertainties in stratospheric $CH_4$ in comparisons of modeled and TCCON observed $XCH_4$. Modeled stratospheric $CH_4$ fields were substituted by satellite-retrieved $CH_4$ fields from MIPAS and ACE-FTS. Original and satellite-corrected model $CH_4$ fields were converted to $XCH_4$ and subsequently evaluated by comparison to TCCON $XCH_4$ observations from 11 sites. This approach and the statistical analysis of $XCH_4$ model-TCCON residuals were conducted with three well-established CTMs: ACTM, TM5 and LMDz.

Our model-TCCON $XCH_4$ intercomparison reveals an inter-model spread in $XCH_4$ bias caused by an inter-model spread in stratospheric $CH_4$. For ACTM we find a large average $XCH_4$ bias of 38.1 ppb, in contrast to small average $XCH_4$ biases of 8.7 ppb for TM5 and 6.8 ppb for LMDz. The ACTM $XCH_4$ bias is reduced by the model correction to 13.7 ppb with MIPAS, and to 3.3 ppb with MIPAS adjusted to ACE-FTS, respectively. For TM5 and LMDz the impact of the model correction with satellite-based $CH_4$ fields is ambiguous. In that. the model $XCH_4$ bias can be slightly reduced to 4.3 ppb with MIPAS, but can also be increased to 10.8 ppb for TM5 and 20.0 ppb for LMDz with MIPAS adjusted to ACE-FTS. This implies that for TM5 and LMDz the model representation of stratospheric $CH_4$ is located within the satellite data range mapped by MIPAS and ACE-FTS observations.

Possible causes for the inter-model spread in stratospheric $CH_4$ have been discussed with an emphasis on model transport characteristics. Applying tropical transport diagnostics suggests that the poor representation of stratospheric $CH_4$ by ACTM originates from errors in the simulation of transport pathways into and within the stratosphere. However, this only is an interpretation based on a diagnostic and requires more process-oriented model evaluation of stratospheric transport. The



inter-model spread in stratospheric $CH_4$ could be quantitatively investigated with a main focus on model-to-model differences in the simulation of stratospheric transport (physical parameterizations, reanalysis data sets, vertical/horizontal resolution), e.g., model simulations could be performed with different reanalysis data sets, and/or different physical parameterizations resulting in a model ensemble for each CTM or a multi-model ensemble consisting of multiple CTM data

sets. This would allow the individual model errors in stratospheric $CH_4$ to be assessed more precisely.

Overall we state that there is a need for improvement in modeling of stratospheric $CH_4$ and, thus, $XCH_4$. At the same time, a better quantification of model errors in stratospheric $CH_4$ is limited by the uncertainty of satellite data products as used in this study. This implies that more stratospheric $CH_4$ in situ observations are required to validate both satellite-retrieved and modeled $CH_4$ data. A more accurate evaluation of modeled stratospheric $CH_4$ fields is particularly reasonable as these CTMs

are used to invert $CH_4$ emissions from $XCH_4$ data. As surface emission signals in $XCH_4$ are small compared to co-resident $XCH_4$ atmospheric background levels, it is necessary to identify minor $XCH_4$ biases in the model as done in this study. Of course, an analogous quality requirement also is needed for ground-based and satellite-borne $XCH_4$ data. Indeed, as long as unallocated and poorly understood differences of several ppb remain between satellite-borne $XCH_4$ data and optimized model fields, it is difficult to take a full benefit of satellite $XCH_4$ data to robustly retrieve regional methane emissions.

**Acknowledgements**

We thank H. P. Schmid (KIT/IMK-IFU) for his continual interest in this work. Our work has been performed as part of the ESA GHG-cci project via subcontract with the University of Bremen. In addition we acknowledge funding by the EC within the InGOS project. A part of work at JAXA was supported by the Environment Research and Technology Development Fund (A-1102) of the Ministry of the Environment, Japan. From 2004 to 2011 the Lauder TCCON program was funded by

the New Zealand Foundation of Research Science and Technology contracts CO1X0204, CO1X0703 and CO1X0406. Since 2011 the program has been funded by NIWA's Atmosphere Research Program 3 (2011/13 Statement of Corporate Intent). The Darwin and Wollongong TCCON sites are funded by NASA grants NAG5-12247 and NNG05-GD07G, and Australian Research Council grants DP140101552, DP110103118, DP0879468, LE0668470, and LP0562346. We are grateful to the DOE ARM program for technical support at the Darwin TCCON site. The Białystok and Orléans TCCON sites are funded

by the EU projects InGOS and ICOS-INWIRE, and by the Senate of Bremen. Nicholas Deutscher was supported by an Australian Research Council fellowship, DE140100178. We also are grateful to P. O. Wennberg for providing TCCON data. The Atmospheric Chemistry Experiment (ACE), also known as SCISAT, is a Canadian-led mission mainly supported by the Canadian Space Agency and the Natural Sciences and Engineering Research Council of Canada.



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





**Figure 1.** Mean CH$_4$ differences between collocated MIPAS and ACE-FTS CH$_4$ profiles measured in the year 2010. Mean CH$_4$ differences in parts per billion (ppb) are derived for 30° latitudinal bands indicated by different colours.






**Figure 2.** Site-specific model XCH$_4$ biases with respect to TCCON observations in parts per billion (ppb) for the year 2010. Different colors indicate different stratospheric CH$_4$ fields used for the calculation of model XCH$_4$.





**Figure 3.** Model-MIPAS differences of stratospheric $CH_4$ volume mixing ratios (vmr) in parts per billion (ppb). Zonally-averaged $CH_4$ vmr differences are annual means for the year 2010.



**Figure 4.** Model-MIPAS_ACE differences of stratospheric CH$_4$ volume mixing ratios (vmr) in parts per billion (ppb). Zonally-averaged CH$_4$ vmr differences are annual means for the year 2010.







**Figure 5.** Model-MIPAS differences of mean age for the tropical lower. Mean age data in years (yr) are calculated as annual means on the MIPAS pressure-latitude grid.



**Figure 6.** Model-MIPAS differences of the mean age gradient as a transport diagnostics for tropical ascent. The mean age gradient was calculated as difference between the lower stratospheric mean ages averaged over 35°N–50°N and 10°S–10°N. Mean age data in years (yr) are calculated as annual means on the MIPAS pressure-latitude grid.



**Figure 7.** Model-MIPAS differences of tropical CH$_4$ mixing ratios as a transport diagnostics for horizontal mixing. The CH$_4$ differences are calculated as annual means on the MIPAS pressure-latitude grid.



**Table 1.** Overview of CTMs used for model-TCCON comparison

| Model name | Institution | Resolution | | Output | Mean age derived from | Reference |
|---|---|---|---|---|---|---|
| | | horizontal[a] | vertical[b] | $CH_4$ | | |
| ACTM | JAMSTEC | ~2.8 × 2.8 ° | 67σ | 1-hourly, monthly | idealized transport tracer simulations | Patra et al. (2016) |
| TM5 | SRON | ~6 × 4 ° | 25η | daily | $SF_6$ simulations | Pandey et al. (2016) |
| LMDz | LSCE | ~3.75 × 1.875 ° | 39η | monthly | $SF_6$ simulations | Locatelli et al. (2015) |

[a] Longitude × Latitude

[b] vertical coordinates in sigma-pressure σ (pressure divided by surface pressure) and hybrid sigma-pressure η



**Table 2.** Overview of TCCON measurement sites used for the evaluation of chemical transport models. Abbreviations of the site names, information about geographical location, and number of measurement days in 2010 are provided.

| TCCON site | Abbreviation | Latitude | Longitude | Days | Reference |
|---|---|---|---|---|---|
| Sodankylä (Finland) | SOD | 67.4 °N | 26.6 °E | 78 | Kivi et al. (2014) |
| Białystok (Poland) | BIA | 53.2 °N | 23.0 °E | 120 | Deutscher et al. (2014) |
| Karlsruhe (Germany) | KAR | 49.1 °N | 8.4 °E | 79 | Hase et al. (2014) |
| Orléans (France) | ORL | 48.0 °N | 2.1 °E | 91 | Warneke et al. (2014) |
| Garmisch (Germany) | GAR | 47.5 °N | 11.1 °E | 120 | Sussmann et al. (2014) |
| Park Falls (USA) | PAR | 46.0 °N | 90.3 °W | 155 | Wennberg et al. (2014a) |
| Lamont (USA) | LAM | 36.6 °N | 97.5 °W | 299 | Wennberg et al. (2014b) |
| Izaña (Tenerife) | IZA | 28.3 °N | 16.5 °W | 50 | Blumenstock et al. (2014) |
| Darwin (Australia) | DAR | 12.4 °S | 130.9 °E | 64 | Griffith et al. (2014a) |
| Wollongong (Australia) | WOL | 34.4 °S | 150.9 °E | 142 | Griffith et al. (2014b) |
| Lauder (New Zealand) | LAU | 45.0 °S | 169.7 °E | 142 | Sherlock et al. (2014a, b) |





**Table 3.** Site-specific model XCH$_4$ biases with respect to TCCON observations in 2010. The model-TCCON agreement in XCH$_4$ is evaluated with different stratospheric CH$_4$ model fields: the original model distribution (**orig**), the MIPAS-based stratospheric CH$_4$ (**MIPAS**), and the MIPAS-based stratospheric CH$_4$ ajusted to ACE-FTS observations (**MIPAS_ACE**). XCH$_4$ biases and corresponding 2-σ standard errors (in brackets) are in parts per billion (ppb).

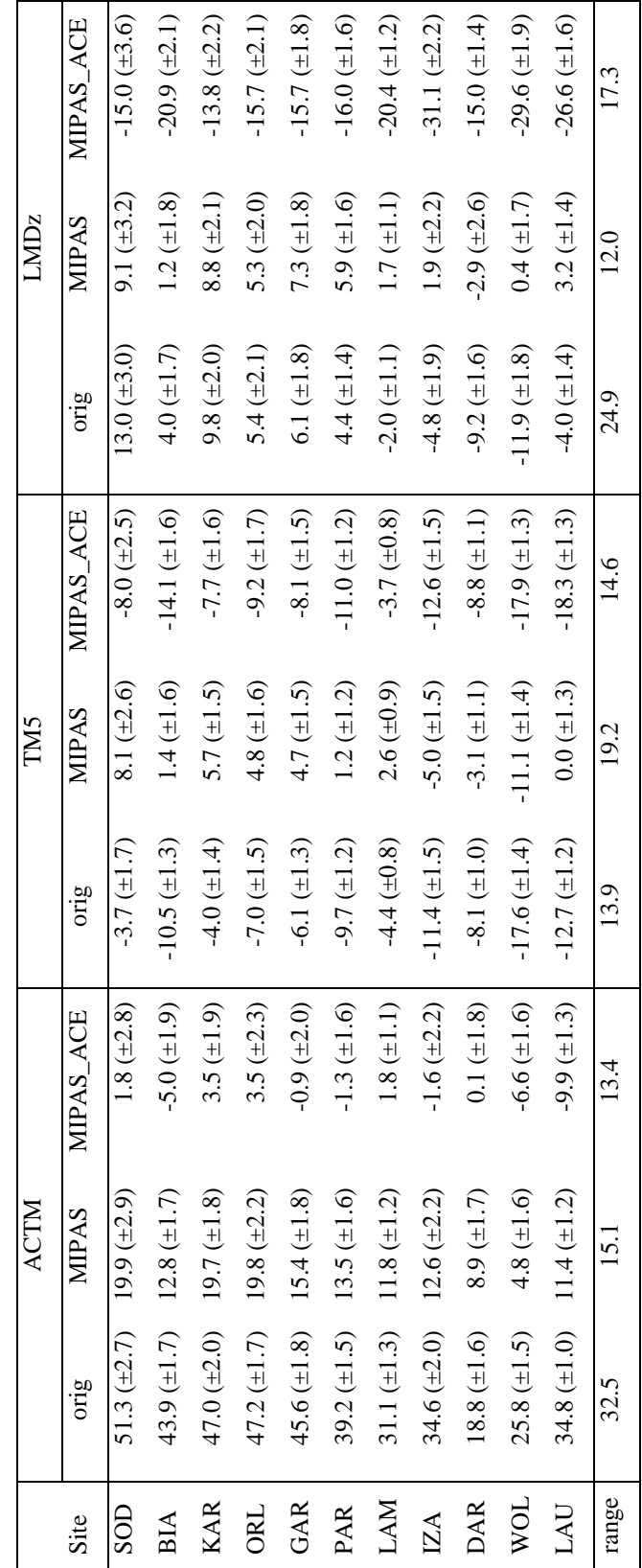

| Site | ACTM | | | TM5 | | | LMDz | | |
|---|---|---|---|---|---|---|---|---|---|
| | orig | MIPAS | MIPAS_ACE | orig | MIPAS | MIPAS_ACE | orig | MIPAS | MIPAS_ACE |
| SOD | 51.3 (±2.7) | 19.9 (±2.9) | 1.8 (±2.8) | -3.7 (±1.7) | 8.1 (±2.6) | -8.0 (±2.5) | 13.0 (±3.0) | 9.1 (±3.2) | -15.0 (±3.6) |
| BIA | 43.9 (±1.7) | 12.8 (±1.7) | -5.0 (±1.9) | -10.5 (±1.3) | 1.4 (±1.6) | -14.1 (±1.6) | 4.0 (±1.7) | 1.2 (±1.8) | -20.9 (±2.1) |
| KAR | 47.0 (±2.0) | 19.7 (±1.8) | 3.5 (±1.9) | -4.0 (±1.4) | 5.7 (±1.5) | -7.7 (±1.6) | 9.8 (±2.0) | 8.8 (±2.1) | -13.8 (±2.2) |
| ORL | 47.2 (±1.7) | 19.8 (±2.2) | 3.5 (±2.3) | -7.0 (±1.5) | 4.8 (±1.6) | -9.2 (±1.7) | 5.4 (±2.1) | 5.3 (±2.0) | -15.7 (±2.1) |
| GAR | 45.6 (±1.8) | 15.4 (±1.8) | -0.9 (±2.0) | -6.1 (±1.3) | 4.7 (±1.5) | -8.1 (±1.5) | 6.1 (±1.8) | 7.3 (±1.8) | -15.7 (±1.8) |
| PAR | 39.2 (±1.5) | 13.5 (±1.6) | -1.3 (±1.6) | -9.7 (±1.2) | 1.2 (±1.2) | -11.0 (±1.2) | 4.4 (±1.4) | 5.9 (±1.6) | -16.0 (±1.6) |
| LAM | 31.1 (±1.3) | 11.8 (±1.2) | 1.8 (±1.1) | -4.4 (±0.8) | 2.6 (±0.9) | -3.7 (±0.8) | -2.0 (±1.1) | 1.7 (±1.1) | -20.4 (±1.2) |
| IZA | 34.6 (±2.0) | 12.6 (±2.2) | -1.6 (±2.2) | -11.4 (±1.5) | -5.0 (±1.5) | -12.6 (±1.5) | -4.8 (±1.9) | 1.9 (±2.2) | -31.1 (±2.2) |
| DAR | 18.8 (±1.6) | 8.9 (±1.7) | 0.1 (±1.8) | -8.1 (±1.0) | -3.1 (±1.1) | -8.8 (±1.1) | -9.2 (±1.6) | -2.9 (±2.6) | -15.0 (±1.4) |
| WOL | 25.8 (±1.5) | 4.8 (±1.6) | -6.6 (±1.6) | -17.6 (±1.4) | -11.1 (±1.4) | -17.9 (±1.3) | -11.9 (±1.8) | 0.4 (±1.7) | -29.6 (±1.9) |
| LAU | 34.8 (±1.0) | 11.4 (±1.2) | -9.9 (±1.3) | -12.7 (±1.2) | 0.0 (±1.3) | -18.3 (±1.3) | -4.0 (±1.4) | 3.2 (±1.4) | -26.6 (±1.6) |
| range | 32.5 | 15.1 | 13.4 | 13.9 | 19.2 | 14.6 | 24.9 | 12.0 | 17.3 |





**Table 4.** Average model XCH$_4$ bias with respect to TCCON observations in 2010 computed as mean of absolute site-specific biases (see Table 3). Average XCH$_4$ biases in ppb are derived for different model stratospheric CH$_4$ fields.

| Model stratospheric CH$_4$ field | mean XCH$_4$ bias | | |
|---|---|---|---|
| | ACTM | TM5 | LMDz |
| Original model | 38.1 | 8.7 | 6.8 |
| MIPAS | 13.7 | 4.3 | 4.3 |
| MIPAS_ACE | 3.3 | 10.8 | 20.0 |