# Peer review of "Model ─ TCCON comparisons of column-averaged methane with a focus on the stratosphere"

_Atmospheric Measurement Techniques, 2016_

## Referee Comment (RC1) · Anonymous Referee #2 · 27 May 2016

[Figure]

**Atmospheric Measurement Techniques Discussions**

The manuscript by Ostler et al. examines the how errors in stratospheric CH4 distributions affect XCH4 (the column-average mole fraction of CH4). The motivation is that inversion analyses often adjust surface emissions to match observed XCH4, but those emission estimates would be wrong if the model's XCH4 error originates in the stratosphere. Ostler et al. find that 3 current models do indeed have sufficiently large errors in stratospheric CH4 that XCH4 is altered by 5-40 ppb, with a systematic latitudinal structure which is large enough to impact emission estimates at a meaningful level. Differences among current stratospheric CH4 observations from satellites imply about 5-10 ppb uncertainty in XCH4, which will likely require more in situ stratospheric measurements to reduce further.

The methods are sound; the figures and analysis are good; and the paper is generally well written. I have a significant criticism of the analysis behind Fig 7, but this is a

[Figure]

secondary issue that does not affect the main analysis. I think this paper deserves publication after addressing the issues below.

Figure 7 attempts to derive stratospheric mixing rates (between tropics and mid latitudes) from the CH4 vertical profiles in the tropics. A similar method has been established by Strahan et al (2011), whom the authors cite, with N2O profiles instead of CH4. N2O has no loss in the lower stratosphere, so the vertical gradients of N2O in the lower tropical stratosphere is due mainly to mixing with low-N2O air in the higher latitudes. Ostler et al. attempt the same technique with CH4, but CH4 does have a significant chemical sink in the lower stratosphere, so the assumption underpinning the technique is violated. I suspect that is why the mixing rates suggested in Fig 7 are at odds with the mean age and ascent rates as described further below. Because the analysis is flawed, I believe Fig 7 needs to be cut. If the authors have N2O simulations and observations, they could use those as a better diagnostic of mixing rates.

Figs 6 and 7 are not entirely consistent with Fig 5. Fig 5 shows that all 3 models have very similar vertical profiles of mean age in the tropics. Fig 6 shows that TM5 has faster vertical ascent in the tropics than the other models, so it should also have greater horizontal mixing between the tropics and mid-latitudes in order to achieve the same mean age as the other models. However, Fig 7 suggests that horizontal mixing in TM5 is not any faster than the other models. I suspect that the use of CH4 instead of N2O as a mixing diagnostic may contribute to this inconsistency.

The MIPAS measurements are averaged for each month, then used as "truth" to replace the model stratosphere fields for comparison to TCCON on individual days. During a month, the tropopause will move up and down in altitude, especially near mid-latitude and subtropical jet streams, which drives a significant change in XCH4 since CH4 mole fractions are generally higher in the troposphere than in the stratosphere. As a result, the stratospheric partial column of CH4 observed by MIPAS will not be correct for the particular days on which TCCON observations are available. The authors mention this issue very briefly but make no attempt to quantify it. I believe it deserves

greater scrutiny, or better explanation of why it is minor compared with other issues.

Clarity issues:

Title: The hyphen in the title can be misinterpreted as meaning that everything after it is clarifying "Model". To avoid any ambiguity I suggest something unambiguous, such as, "Evaluation of column-averaged methane in models and TCCON with a focus on the stratosphere".

On Page 1 Line 30 (P1L30), it is not clear that the model-TCCON agreement is improved by *substituting* the MIPAS-based stratospheric CH4 observations *in place of* the model's stratospheric CH4 simulation. Similarly on line 34, it's not clear that the simulated stratospheric CH4 is again replaced with a different satellite CH4 product.

P1L33: "respectively" is not needed.

P1L35: "These findings imply..." sentence is not clear to me. I think it contains two claims: "These findings imply that model errors in simulating stratospheric CH4 contribute to model biases" and "Current satellite instruments cannot definitively measure stratospheric CH4 to sufficient accuracy to eliminate these biases."

P2L5: The stratospheric chemistry community has devoted a lot of time, research, and papers to understanding these issues. Some of those papers are cited in this work, but a great many are not. Perhaps the specific models used in this work have not been part of those studies, but it seems over broad to say that the these issues haven't been studied adequately.

P2L33: What is a "residual bias"? Residual after doing what and compared to what?

P3L1: I believe there are too many negatives (cannot, without, unambiguous), e.g. "without" should be "with".

P3L2: What is a "bias function"?

P10L4: Check sentence grammar.

---

## Referee Comment (RC2) · C. Frankenberg (Referee) · 4 Aug 2016

First of all, my sincere apologies for the late review, there is no excuse for this.

The paper by Ostler et al deals with the impact of stratospheric CH4 on model TCCON comparisons. Given the relative importance of stratospheric methane on global flux inversions, the paper warrants publication. I find it generally well written and very suitable for ACP. I have a few comments that might help to improve the paper and make some aspects a bit more general and not too confined to TCCON comparisons only.

In general, I am not sure whether mean bias is really the best metric to use for quantifying "success", esp. as all satellite data might have a small residual bias, which can be scaled to optimize agreement (also holds for TCCON, SCIA, GOSAT and stratospheric

data). I would consider the station to station bias variability (similar to the range used by the authors) as well as the ability to capture seasonality a better metric (seasonality not quantified here). Most inversions will include a general bias correction term anyhow. Station elevation would also be an important aspect as it determines the fractional contribution of the stratosphere to XCH4 (add in Table 2, probably only important for Izana). This also changes with seasons, so a look at whether the method here improves the seasonality of models would be very worthwhile looking into. Figure 2: It would be good to also look at the latitudinal difference in a more general sense, e.g. a global average and spread of the differences as opposed to just at TCCON stations. If the global difference fields have already been computed, it would be very easy to do so but I am not sure whether this was done. This could be a valuable addition to the paper as it will increase the relevance to flux inversions. A separate DJF and JJA plot would also be good to reflect the impact on seasonality as well. Figs 3/4: While it is customary to show stratospheric variables in a log P scale, I would find a linear y-axis in pressure more useful in this case as it enable the reader to better estimate the impact on column values. Right now, the eye might often be focussed on some of the strong variations at lower p (e.g around 10hPa), which might be striking but could be irrelevant for the column integral.

One other question is how the measured fields are replaced in the models. It is stated that it can sometimes even be in the troposphere. Is it a brute force replacement (i.e. will there be discontinuities in the updated model field?). What happens if you define a transition range in p where you "smoothly" replace the model with the updated fields? Would it matter?

---

## Author Comment (AC1) · 12 Sep 2016

**Final Response, Andreas Ostler, Karlsruhe Institute of Technology, Garmisch-Partenkirchen, Germany, 10 September 2016**
It is a pleasure to thank both the referee Christian Frankenberg and the Anonymous Referee for very sound and helpful comments which lead to significant improvements and interesting extensions of the paper. We thereafter present our point to point reply.

**Referee #2:**

*I have a significant criticism of the analysis behind Fig 7, but this is a secondary issue that does not affect the main analysis.*
*Figure 7 attempts to derive stratospheric mixing rates (between tropics and mid latitudes) from the CH4 vertical profiles in the tropics. A similar method has been established by Strahan et al (2011), whom the authors cite, with N2O profiles instead of CH4. N2O has no loss in the lower stratosphere, so the vertical gradients of N2O in the lower tropical stratosphere is due mainly to mixing with low-N2O air in the higher latitudes. Ostler et al. attempt the same technique with CH4, but CH4 does have a significant chemical sink in the lower stratosphere, so the assumption underpinning the technique is violated. I suspect that is why the mixing rates suggested in Fig 7 are at odds with the mean age and ascent rates as described further below. Because the analysis is flawed, I believe Fig 7 needs to be cut. If the authors have N2O simulations and observations, they could use those as a better diagnostic of mixing rates.*

We agree that the use of $CH_4$ instead of $N_2O$ as a mixing diagnostic may lead to inconsistencies. Therefore, and since we don´t have $N_2O$ simulations and observation at hand, we agree to cut out Fig. 7. Consequently, we removed all text passages to Fig. 7.

*The MIPAS measurements are averaged for each month, then used as "truth" to replace the model stratosphere fields for comparison to TCCON on individual days. During a month, the tropopause will move up and down in altitude, especially near midlatitude and subtropical jet streams, which drives a significant change in XCH4 since CH4 mole fractions are generally higher in the troposphere than in the stratosphere. As a result, the stratospheric partial column of CH4 observed by MIPAS will not be correct for the particular days on which TCCON observations are available. The authors mention this issue very briefly but make no attempt to quantify it. I believe it deserves greater scrutiny, or better explanation of why it is minor compared with other issues.*

We agree that vertical shifts of the tropopause can cause significant changes in $XCH_4$. In Ostler et al. (2014) we showed that $XCH_4$ variations of ~25 ppb even can occur within a day. As you wrote, the tropopause can be shifted upwards and downwards, i.e. the $XCH_4$ changes can be positive but also negative. Consequently, we expect that these dynamically induced $XCH_4$ variations should be negligible in a statistical point of view as used in this study. In order to clarify this, we added the following sentences at Sect. 3.3: "E.g. vertical shifts of the tropopause can cause significant variations in $XCH_4$ of ~25 ppb even within a day (Ostler et al., 2014). As these $XCH_4$ changes can be positive but also negative (tropopause shifted upwards and downwards), we expect that dynamically induced $XCH_4$ variations should be negligible in a statistical point of view as used in this study."

*Clarity issues:*

*Title: The hyphen in the title can be misinterpreted as meaning that everything after it is clarifying "Model". To avoid any ambiguity I suggest something unambiguous, such as, "Evaluation of column-averaged methane in models and TCCON with a focus on the stratosphere".*

We changed the title according to your suggestion.

*On Page 1 Line 30 (P1L30), it is not clear that the model-TCCON agreement is improved by *substituting* the MIPAS-based stratospheric CH4 observations *in place of* the model's stratospheric CH4 simulation. Similarly on line 34, it's not clear that the simulated stratospheric CH4 is again replaced with a different satellite CH4 product.*

To clarify, we changed the wording as follows:
Using MIPAS-based stratospheric $CH_4$ fields in place of model simulations improves the model-TCCON $XCH_4$ agreement for all models.
Replacing model simulations with MIPAS stratospheric $CH_4$ fields adjusted to ACE-FTS reduces the average $XCH_4$ bias.

*P1L33: "respectively" is not needed.*

The word "respectively" was removed.

*P1L35: "These findings imply: : :" sentence is not clear to me. I think it contains two claims: "These findings imply that model errors in simulating stratospheric CH4 contribute to model biases" and "Current satellite instruments cannot definitively measure stratospheric CH4 to sufficient accuracy to eliminate these biases."*

We agree that the formulation of this sentence is not adequate. Hence, the original sentence was replaced by your suggestions.

*P2L5: The stratospheric chemistry community has devoted a lot of time, research, and papers to understanding these issues. Some of those papers are cited in this work, but a great many are not. Perhaps the specific models used in this work have not been part of those studies, but it seems over broad to say that these issues haven't been studied adequately.*

We agree to the point and changed the wording as follows:
"Therefore, it would be worthwhile to analyze how individual model components (e.g., physical parameterization, meteorological data sets, model horizontal/vertical resolution) impact the simulation of stratospheric $CH_4$ and $XCH_4$."

*P2L33: What is a "residual bias"? Residual after doing what and compared to what?*

As stated in Houweling et al. (2014), well-quantified biases are usually directly corrected in the model or the measurements. Nevertheless, there are random and systematic errors that are remaining after such corrections. These remaining biases also are residual biases.
As the word "residual" may lead to confusion at this point, it has been removed.

*P3L1: I believe there are too many negatives (cannot, without, unambiguous), e.g. "without" should be "with".*

We agree and changed the wording to "… can only be attributed to errors in the model or the observations with an ambiguous assignment."

*P3L2: What is a "bias function"?*

Unfortunately, there was a word missing here which we inserted to the revised manuscript: It should be "bias correction function". These correction functions are based on model-observation differences and can depend on season and latitude. More details can be found in Sect. 2.4 of Houweling et al. (2014).

***P10L4: Check sentence grammar.***

Changed.

**Referee #1 (C. Frankenberg):**

*In general, I am not sure whether mean bias is really the best metric to use for quantifying "success", esp. as all satellite data might have a small residual bias, which can be scaled to optimize agreement (also holds for TCCON, SCIA, GOSAT and stratospheric data). I would consider the station to station bias variability (similar to the range used by the authors) as well as the ability to capture seasonality a better metric (seasonality not quantified here). Most inversions will include a general bias correction term anyhow. Station elevation would also be an important aspect as it determines the fractional contribution of the stratosphere to XCH4 (add in Table 2, probably only important for Izana). This also changes with seasons, so a look at whether the method here improves the seasonality of models would be very worthwhile looking into.*

Our study uses model, satellite, and TCCON data covering one single year (2010). Given this relatively small time period we did not consider to analyze seasonality effects in addition to the basic metric mean bias. While we agree that investigation of seasonality effects would be highly interesting this would require using a significantly larger data set. This is beyond the scope of this paper but could well be topic of a subsequent study.

Furthermore, we added station elevation in Table 2 as suggested, and briefly referred to station elevation effects in the text (Sect. 4).

*Figure 2: It would be good to also look at the latitudinal difference in a more general sense, e.g. a global average and spread of the differences as opposed to just at TCCON stations. If the global difference fields have already been computed, it would be very easy to do so but I am not sure whether this was done. This could be a valuable addition to the paper as it will increase the relevance to flux inversions. A separate DJF and JJA plot would also be good to reflect the impact on seasonality as well.*

We extended Section 4 and added a part where latitudinal $XCH_4$ differences between original and satellite-corrected model fields are shown in a new Figure 5 (including separate DJF and JJA plots).

*Figs 3/4: While it is customary to show stratospheric variables in a log P scale, I would find a linear y-axis in pressure more useful in this case as it enable the reader to better estimate the impact on column values. Right now, the eye might often be focused on some of the strong variations at lower p (e.g around 10hPa), which might be striking but could be irrelevant for the column integral.*

The axes in Figs. 3 and 4 have been changed to linear scale.

*One other question is how the measured fields are replaced in the models. It is stated that it can sometimes even be in the troposphere. Is it a brute force replacement (i.e. will there be discontinuities in the updated model field?). What happens if you define a transition range in p where you "smoothly" replace the model with the updated fields? Would it matter?*

Yes, it is a force replacement with some discontinuities in the updated model field. To address your point, we now also performed a smooth replacement using a transition range between model and satellite field. The method and the results are described in the revised Sect. 4. The main result is that the impact of a smoothed replacement in terms of $XCH_4$ is relatively small compared to the force replacement (see new Fig. 6).

**Extension of Section 4:**

The zonal difference fields between model and satellite-based $CH_4$ data sets have also been converted to $XCH_4$ differences and are shown in Fig. 5. Two main features can be found in Fig. 5: (*i*) the $XCH_4$ difference range between the two satellite-based data sets MIPAS (dark red) and MIPAS_ACE (light red), which is ~27 ppb (1-σ *stdv*=4 ppb) on annual mean basis. (*ii*) Model-satellite $XCH_4$ differences indicating a latitudinal dependence for ACTM (Fig. 1a) and LMDz (Fig. 1c). E.g. ACTM-satellite $XCH_4$ differences clearly are increasing toward higher latitudes. In contrast to this, the TM5-satellite $XCH_4$ difference does not show a latitudinal dependence. These findings on the latitudinal dependence of model-satellite $XCH_4$ differences are supported by Table 5 providing some statistical results. E.g. the *stdv*s and the min-max ranges of model-satellite $XCH_4$ differences are much smaller for TM5 compared to the other models. Besides that, Fig. 5 also shows that the model-satellite $XCH_4$ differences for the year 2010 only slightly depend on season. A noticeable seasonal variation in the model-satellite $XCH_4$ differences can be found in the tropical/subtropical region of the Northern Hemisphere. However, in order to analyze seasonal variations, a more thorough analysis is needed including model and satellite-based $XCH_4$ data sets with a larger time period than used in this study. Furthermore, in the context of seasonality the role of TCCON station elevation needs to be considered in more detail. Since we only apply one year of model and satellite data, the focus of this study is not on the seasonal agreement between model and satellite-based $XCH_4$ data sets.

Modeled stratospheric $CH_4$ fields have been directly replaced by satellite data sets. As a result, there can be discontinuities in the merged $CH_4$ fields around the tropopause, where the lowest satellite-based $CH_4$ mixing ratios strongly deviate from the original modeled $CH_4$ mixing ratios. In order to quantify the impact of these discontinuities on the $XCH_4$ data sets, we also have performed a smoother replacement method. For this purpose we defined a vertical transition range of 75 hPa starting at the lowest vertical MIPAS data grid point. From this position the model vertical profile of $CH_4$ mixing ratios linearly was interpolated to the satellite-based $CH_4$ mixing ratio profile starting at the upper boundary of this transition range. This method was applied to each latitudinal MIPAS grid point corresponding to a vertical profile of $CH_4$ mixing ratios. The method was not used, if the model-satellite difference of $CH_4$ mixing ratios was smaller than 30 ppb at the lower boundary of the transition range. Consequently, we also computed $XCH_4$ differences between the original model and the smoothed satellite-based data sets. Figure 6 then shows model-satellite $XCH_4$ differences resulting from the force replacement (solid lines) and from the smoothed replacement (dashed lines). From Fig. 6 it is obvious that the impact of the smoothing replacement on the model-satellite $XCH_4$ differences is small, i.e. differences between solid and dashed lines typically are smaller than 4 ppb. For this reason we expect that the impact of discontinuities in the merged model-satellite $CH_4$ fields on the results of the $XCH_4$ validation against TCCON is negligible.

[Figure]

**Figure 5.** Zonal XCH$_4$ differences resulting from model-satellite differences of stratospheric CH$_4$ volume mixing ratios. Mean XCH$_4$ differences are shown as solid lines for the summer period (June, July and August) and as dashed lines for the winter period (December, January and February).

[Figure]

**Figure 6.** Zonal XCH$_4$ differences as a result of model-satellite differences of stratospheric CH$_4$ volume mixing ratios. Solid lines refer to the merged model-satellite CH$_4$ fields including discontinuities at the model-satellite transition zone around the tropopause. Dashed lines refer to merged model-satellite CH$_4$ fields that have been smoothed at the model-satellite transition zone.

**Table 5 Average XCH$_4$ differences between model simulations and model CH$_4$ fields with satellite-based stratospheric CH$_4$ fields. Annual mean difference as XCH$_4$ bias (with 1-σ *stdv*) and minimum-maximum range of zonal XCH$_4$ differences are in ppb.**

| Satellite data | ACTM | | TM5 | | LMDz | |
|---|---|---|---|---|---|---|
| | bias | min-max | bias | min-max | Bias | min-max |
| MIPAS | 22.3 (±14.1) | 45.2 | −13.9 (±3.4) | 12.8 | −4.3 (±9.4) | 29.3 |
| MIPAS_ACE | 48.7 (±11.0) | 35.4 | 13.6 (±3.5) | 14.8 | 23.2 (±6.8) | 22.3 |

References:

Houweling, S., Krol, M., Bergamaschi, P., Frankenberg, C., Dlugokencky, E. J., Morino, I., Notholt, J., Sherlock, V.,Wunch, D., Beck, V., Gerbig, C., Chen, H., Kort, E. A., Röckmann, T., and Aben, I.: A multi-year methane inversion using SCIAMACHY, accounting for systematic errors using TCCON measurements, Atmos. Chem. Phys., 14, 3991–4012, doi:10.5194/acp-14-3991-2014, 2014.

Ostler, A., Sussmann, R., Rettinger, M., Deutscher, N. M., Dohe, S., Hase, F., Jones, N., Palm, M., and Sinnhuber, B.-M.: Multistation intercomparison of column-averaged methane from NDACC and TCCON: impact of dynamical variability, Atmos. Meas. Tech., 7, 4081-4101, doi:10.5194/amt-7-4081-2014, 2014.